# Learning System Dynamics from Sensory Input under Optimal Control Principles

## Abstract

Identifying the underlying dynamics of actuated physical systems from sensory input is of high interest in control, robotics, and engineering in general. In the context of control problems, existing approaches decouple the construction of the feature space where the dynamics identification process occurs from the target control tasks, potentially leading to a mismatch between feature and real state spaces: the systems may not be controllable in feature space, and synthesized controls may not be applicable in the state space. Borrowing from the Koopman formalism, we propose instead to learn an embedding of both the states and controls in feature spaces where the dynamics are linear, and to include the target control task in the learning objective in the form of a differentiable and robust optimal control problem. We validate this approach with simulation experiments of systems with non-linear dynamics, demonstrating that the controls obtained in feature space can be used to drive the corresponding physical systems and that the learned model can serve for future state prediction.

## 1 Introduction

The study of dynamical systems is a key element in understanding most physical phenomena. Such systems are ruled by ordinary differential equations of state variables that contain enough information to describe and determine their behavior, and analytical models of these systems are traditionally derived as solutions of the differential equations in question. However, it is hard to fully model mathematically most real-life phenomena for several reasons: they may have very complex dynamics with complex and constantly changing interactions with the environment, and the state of the physical systems involved may be unknown or not fully observable. On the other hand, the physical systems themselves, if not their internal states, can be observed with sensory data providing implicit information about the underlying (and unknown) states. Thus, leveraging measurement data is natural, and is actually done in a wide range of approaches which build representations of systems from past measurements in the form of feature spaces (Brunton et al., 2016b; Arbabi et al., 2018; Bruder et al., 2019; Brunton et al., 2021). These models are of high practical interest since they enable compact representations compared to the density of measurements (e.g., when measurements are images). They also enable lifting the state of the system to a higher dimensional space where predictive models can be built. However, even when effective, these estimated models and feature spaces remain highly uninterpretable, and using them in solving control problems remains challenging. Linear models on the contrary are easily interpretable, and enable exact and effective control when coupled with LQR solvers. In particular, the Koopman operator theory (Koopman, 1931) has gained a lot of interest recently (Proctor et al., 2016; Brunton et al., 2016b; Abraham et al., 2017; Morton et al., 2018; Korda & Mezić, 2018; Arbabi et al., 2018; Brunton et al., 2021). It guarantees the existence of a linear (if typically infinite-dimensional) representation of the dynamics of the observables (vector-valued functions) defined over the state space. Finite dimensional approximations have been proposed, and dynamic mode decomposition (DMD) (Schmid, 2010) is of particular interest in this context. In DMD, an approximation of the Perron-Frobenius operator, adjoint to the Koopman operator, is constructed in the form of a matrix that ensures the transition from one observation to the next. Proctor et al. (2016) first extended the use of DMD to actuated systems and modeled the system dynamics as a linear function of the state representation and the control. Several works have built upon this approximation (Morton et al., 2018; Li et al., 2020) and various methods for estimating the corresponding operators have been proposed (Morton et al.,

2018; Xiao et al., 2021; Li et al., 2020). In all these works, these operators are constructed to solve a prediction task, assuming the controls are known and, once obtained, they are used in a control task, typically an LQR problem. However, there are two main issues with this decoupled approach. First, the learned features may not be adapted to the control task, since they were not trained for it. Thus, the modeled dynamics are not guaranteed to be effective when used as (linear) constraints to minimize a given (quadratic) cost. We believe that including the control problem in the learning process should help learning features that are well suited for both prediction *and* control. Second, when looking for a couple of matrices that satisfy the desired linearity property in the feature space, it is assumed that the dynamics are linear in the real controls. This is a strong assumption that is not necessarily satisfied, and is not justified by Koopman operator theory. In addition, the dimension of the observations representations is usually larger than that of the (unknwon) states, resulting in a representation that has potentially more degrees of freedom than the real system. Thus, if the dimension of the control in feature space remains that of the original system, the system might not be controllable anymore. This is justified by the Kalman criterion for controllability (Kalman, 1964) which is that the controllability matrix is full rank. This is facilitated by having observation and control feature spaces of similar dimensions. To address this issue, we propose to also lift the controls to a higher dimensional space in order to avoid rank-deficiency issues in the controllability matrix. To summarize, we learn in this work a representation of dynamical systems from measurements where a linear model of their dynamics can be identified and used to predict their behaviour and control them. In this representation space, both states and controls are lifted to a higher dimension, which makes the system linearly controllable in this space. We are able to do so by directly including a control task in the learning objective, which allows us to learn representations that are better suited for control. We include experiments that show the effectiveness of our approach in controlling pendulum and cartpole systems in simulation.

## 2 METHOD

### 2.1 CONTROLLED DYNAMICAL SYSTEMS

Actuated dynamical systems are dynamical systems whose state $x(t) \in \mathcal{X} \subset \mathbb{R}^n$ follows a differential equation of the form $\dot{x}(t) = f(x(t), u(t))$, where $f$ is a (non-linear) function, and $u(t)$ a control input in a control space $U \subset \mathbb{R}^p$. In this work, we study discrete time actuated dynamical systems, i.e., systems whose discrete state $x_t$ in $\mathcal{X} \subset \mathbb{R}^n$ follows an equation of the type

$$x_{t+1} = f_t(x_t, u_t), \tag{1}$$

where $u_t$ in $\mathcal{U} \subset \mathbb{R}^p$ is a control vector. In practical settings, the model $f_t$ is unknown, and the states $x_t$ are unknown or only partially observable through sensors $g_t$. In this work, we consider measurements $d_t = g_t(x_t)$, and seek to learn encodings of the unknown states of the system from these measurements.

### 2.2 APPROACH

We want to learn encodings of both the states and the controls that have three properties: first, the code for the system state at time $t$, constructed from system measurements at that time $t$, should contain enough information to capture the behavior of the system at that time. Second, we want the dynamics to be linear in codes space, even though they may be arbitrary in the original state space. This is motivated in part by the Koopman linear representation of arbitrary dynamics for non-actuated systems. Third, we want the system to be controllable in the learned representation space. Because of the first and second properties, the system dynamics are lifted to a higher dimensional space. In such a space, the system representation has more degrees of freedom, and might not be controllable with the original controls anymore. In our approach, we propose to also learn an encoding of the controls through a second autoencoder, in order to lift the controls to a higher dimensional space.

**Encoding.** Let us consider a dynamical system governed by Eq. (1). We assume that the states $x_t$ and the model $g$ are unknown, and that we only have access to a sequence of $T$ measurements $d_t$ in $\mathcal{I} \subset \mathbb{R}^{C \times H \times W}$ of the system (images in our case). We want to learn the parameters of encoders $\varphi : \mathcal{I} \to \mathbb{R}^n$ and $\psi : \mathcal{U} \to \mathbb{R}^d$ such that:

$$z_t = \varphi(d_t), \ \ c_t = \psi(u_t), \ \ \text{for } t = 1, \ldots, T. \tag{2}$$

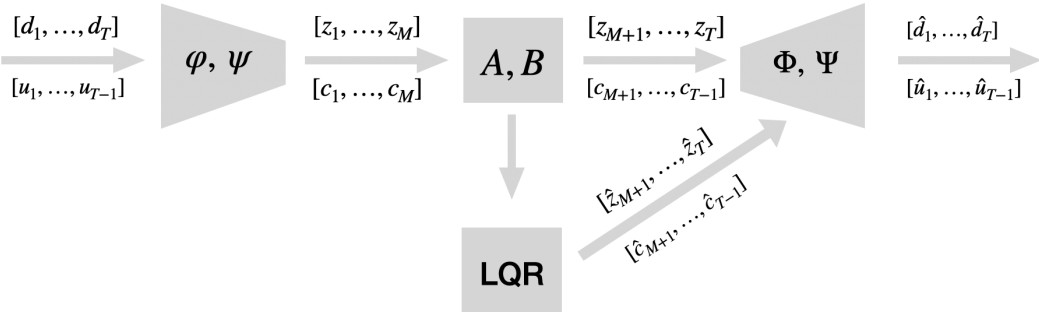

Figure 1: Method. Measurements $d_t$ and controls $u_t$ are input respectively to the image and control encoders. The first $M$ elements are encoded and used to estimate dynamics matrices $A$ and $B$. Codes from $M+1$ to $T$ are obtained in two ways: (1) as solutions to the LQR problem of driving the system from its configuration at time $M$ to its configuration at time $T$ in $T-M$ time steps, and (2) as predictions obtained with the forward linear model associated with the matrices $(A, B)$. All codes $z_t$ and $c_t$ are then input respectively to the image and control decoders, and image and control reconstructions $\hat{d}_t$ and $\hat{u}_t$ are obtained.

The encoders $\varphi, \psi$ are learned in an auto-encoding fashion jointly with their decoder counterparts $\Phi : \mathbb{R}^n \to \mathcal{I}$ and $\Psi : \mathbb{R}^d \to \mathcal{U}$ such that $\hat{d}_t = \Phi(z_t)$ and $\hat{u}_t = \Psi(c_t)$.

**Linear model estimation.** Based on Koopman operator theory, we want the dynamics to be linear in the representation space. We use an approach similar to the DMD approximation to look for dynamics and control matrices $A$ and $B$. Our goal is to learn a representation space where the dynamics are linear, i.e., we want to find matrices $A$ and $B$ such that

$$z_{t+1} = Az_t + Bc_t \quad \text{for all } t. \tag{3}$$

To do so, we split the sequence of $T$ measurements in two subsequences. The first one, from 1 to $M$ (where $M$ is an integer lower than $T$) is used to estimate the matrices $A$ and $B$, and the second one, from $M+1$ to $T$, is used to verify that the dynamics matrices enable prediction. Formally, $A$ and $B$ are estimated using $z_1, \ldots, z_M$ and $c_1, \ldots, c_M$ as

$$(A, B) = \arg \min_{(P,Q)} \sum_{t=1}^{M-1} \|z_{t+1} - Pz_t - Qc_t\|_2^2. \tag{4}$$

To solve this problem, we follow Li et al. (2020) and Bounou et al. (2021): defining the matrices $Z_1 = [z_1 \ldots z_{M-1}]$, $Z_2 = [z_2 \ldots z_M]$, and $C = [c_1 \ldots c_{M-1}]$, we solve

$$(A, B) = \arg \min_{(P,Q)} \|Z_2 - PZ_1 - QC\|_F^2. \tag{5}$$

Because the matrices $Z_1$, $Z_2$ and $C$ are constructed from learned encodings, they are not guaranteed to be full-rank, making the linear least-squares problem (5) ill-posed. To avoid instabilities during training and ensure we always obtain a solution, we use the proximal method of multipliers (Rockafellar, 1976) and solve iteratively the well-posed problem

$$(A^{k+1}, B^{k+1}) = \arg \min_{(P,Q)} \|Z_2 - PZ_1 - QC\|_F^2 + \frac{\rho}{2}\|P - A^k\|_F^2 + \frac{\rho}{2}\|Q - B^k\|_F^2 \tag{6}$$

until convergence, starting from random initial matrices $A^0$ and $B^0$. Unlike other regularization schemes such as adding L2 regularizations to $P$ and $Q$, this iterative procedure is guaranteed to converge to a solution to the original problem (5) (Rockafellar, 1976; Parikh & Boyd, 2014), and not to a solution to a shifted problem as may happen with L2 regularizations.

**Future representation prediction.** The future code values $z_{M+1}, \ldots z_T$ are obtained using the forward linear model:

$$z_{t+1} = Az_t + Bc_t, \quad \text{for } t \text{ in } M, \ldots, T-1. \tag{7}$$

This step ensures that the learned encoders $\varphi, \psi$ and the matrices $A$ and $B$ are suited for prediction.

**LQR control task.** Finally, we want the system to be controllable in codes space. The goal is to drive the system from an initial configuration in the encoding space to a target in that same space. We formalize this as an LQR problem:

$$(\hat{z}, \hat{c}) = \arg\min_{z,c} \sum_{t=M}^{T-1} (z_t - z_T^*)^T Q (z_t - z_T^*) + c_t^T R c_t + (z_T - z_T^*)^T Q_f (z_T - z_T^*) \quad (8a)$$

$$\text{s.t. } z_{t+1} = A z_t + B c_t,$$
$$z_M = \varphi(d_M), \quad (8b)$$
$$z_T^* = \varphi(d_T),$$

where $Q$ and $Q_f$ are symmetric positive semi-definite state cost matrices, and $R$ is a symmetric positive definite control cost matrix. Here, $z_M$ is the initial configuration of the system, and $z_T^*$ is the target configuration we want the system to reach. Both encode measurements of the system in initial and target configurations (e.g., if the system is a pendulum and the measurements are images, $z_M$ and $z_T^*$ are encodings of the images of the pendulum at the original position and at the desired target position.). The quadratic cost in Eq. (8a) has a tracking term that pushes the variables $z_t$ to be close to the target $z_T^*$ all along the control trajectory.

**Supervision.** Equations (2), (7) and (8) lead to three situations: from 1 to $M$, the variables $z_t$ are otbained directly by encoding the corresponding measurements $d_1, \ldots, d_M$. From $M+1$ to $T$, they are obtained by the forward linear model $z_{t+1} = A z_t + B c_t$, using encodings of the known controls $u_{M+1}, \ldots, u_T$. Finally, $\hat{z}_{M+1}, \ldots, \hat{z}_T$ are obtained by solving the LQR problem of driving the system in the encoding space from $\hat{z}_M$ to the target $\hat{z}_T$. There are two types of control features: control encodings $c_1, \ldots c_M$ from Eq. (2), and solutions to the LQR problem $\hat{c}_{M+1}, \ldots, \hat{c}_T$ from problem (8). In all cases, the codes $z_t$ and $c_t$ should match the original measurements once decoded:

$$\hat{d}_1, \ldots, \hat{d}_T = \Phi(z_1), \ldots, \Phi(z_T), \quad (9)$$

$$\hat{u}_1, \ldots, \hat{u}_T = \Psi(c_1), \ldots, \Psi(c_T). \quad (10)$$

**Training objective.** Our model is trained to minimize the empirical risk of the loss:

$$\mathcal{L}_\theta(\{\mathbf{d}^i, \mathbf{u}^i\}_{i=1,\ldots,N}) = \frac{1}{N} \sum_{i=1}^{N} \Big( \sum_{t=1}^{M} \underbrace{\|d_t^i - \Phi(\varphi(d_t^i))\|_2^2}_{\text{Measurements AE loss}} + \sum_{t=M}^{T-1} \underbrace{\|d_{t+1}^i - \Phi(A_i \varphi(d_t^i) - B_i \psi(u_t^i))\|_2^2}_{\text{Prediction loss}}$$

$$+ \sum_{t=0}^{T-1} \underbrace{\|u_t^i - \Psi(\psi(u_t^i))\|_2^2}_{\text{Controls AE loss}} + \sum_{t=M+1}^{T} \underbrace{\|d_t^i - \Phi(\hat{z}_t^i)\|_2^2}_{\text{LQR reco loss}} + \sum_{t=M}^{T-1} \underbrace{\|u_t^i - \Psi(\hat{c}_t^i)\|_2^2}_{\text{LQR controls loss}} \Big),$$

$$(11)$$

where $\theta$ is the set of parameters of the encoders $\varphi$ and $\psi$ and decoders $\Phi$ and $\Psi$. The first term uses the auto-encoder measurements loss and ensures that the encoder learns a representation with enough information about the system at a given timestep to reconstruct the observations. The second one uses a prediction loss which ensures that the estimated matrices $(A, B)$ enable the prediction of future codes in the latent space (that are then decoded to the measurement space). The third term uses the auto-encoder controls loss and ensures that the control encoding can be mapped back to its original space. The last two terms are associated with the control task: they ensure that the decodings of the optimal solutions to the LQR match the original optimal measurements and control trajectories. Since the different terms in the loss are not in the same scale nor the same unit, we add scaling coefficients before each one of them for training.

**LQR formulation with delayed coordinates.** In practice, raw measurement data may not contain sufficient information for future prediction. It is the case for example when we observe a single image of an oscillating pendulum at a given time step and we can not tell from this image whether the pendulum is going up or down. Instead, encodings of at least two consecutive measurements are needed to predict the future code. This case is easily handled using the classical trick of using augmented feature representations associated with multiple frames, similarly to what is done in

Takeishi et al. (2017). Let us define the augmented vector $\tilde{z}_t^{1:h} = \begin{bmatrix} z_{t-h+1}^T & \dots z_t^T \end{bmatrix}^T$, where $h$ is the number of consecutive measurements we choose to consider. The new dynamics equation becomes $z_{t+1} = A^{1:h} z_t^{1:h} + Bc_t$, where $A^{1:h} = [A_1 \quad \dots \quad A_h]$. To be in the formalism of Eq. (8b), we define the augmented matrices $\tilde{A}$ and $\tilde{B}$ as

$$\tilde{A} = \begin{bmatrix} 0 & I & 0 & \dots & \dots & 0 \\ 0 & 0 & I & 0 & \dots & 0 \\ \dots & \dots & \dots & \dots & \dots & \dots \\ A_1 & \dots & \dots & \dots & \dots & A_h \end{bmatrix}, \tilde{B} = \begin{bmatrix} 0 \\ \dots \\ 0 \\ B \end{bmatrix}, \tag{12}$$

where $\tilde{A}$ is of size $(hn \times hn)$ and $\tilde{B}$ is of size $(hn \times d)$. We also define augmented initial and terminal constraints $\tilde{z}_M^{1:h} = \begin{bmatrix} z_{M-h+1}^T & \dots & z_M^T \end{bmatrix}^T$ and $\tilde{z}_T^{:*} = \begin{bmatrix} z_{T-h+1}^T & \dots & z_T^T \end{bmatrix}^T$, both of size $hn$. With these notations, the new dynamics equation becomes:

$$\tilde{z}_{t+1} = \tilde{A}\tilde{z}_t + \tilde{B}c_t. \tag{13}$$

## 2.3 RELATED WORK

Data-driven approaches based on the Koopman operator theory (Koopman, 1931) have gained in popularity in the context of building dynamical models directly from measurements (Morton et al., 2018; Brunton et al., 2016a; Abraham et al., 2017; Arbabi et al., 2018; Bruder et al., 2019; Takeishi et al., 2017; Azencot et al., 2020; Xiao et al., 2021; Lusch et al., 2018; Li et al., 2020). Borrowing from the Perron-Frobenius operator, adjoint to the Koopman operator (Brunton et al., 2021; Klus et al., 2015), these approaches are grounded in the formalism enabled by the dynamic mode decomposition (DMD) (Schmid, 2010). In this context, several approaches have been proposed to construct a representation space with linear dynamics. Such a space can either be handcrafted (Brunton et al., 2016a; Abraham et al., 2017; Arbabi et al., 2018; Bruder et al., 2019) or learned (Takeishi et al., 2017; Lusch et al., 2018; Morton et al., 2018; Azencot et al., 2020; Li et al., 2020; Xiao et al., 2021). In both cases, the representation space is built in the form of a mapping from the measurements space to a latent space. The DMD framework has also been extended to actuated systems for model identification and control purposes in Proctor et al. (2016). Based on this extension, Brunton et al. (2016b); Morton et al. (2018); Arbabi et al. (2018); Bruder et al. (2019); Li et al. (2020) and Bounou et al. (2021) seek to identify dynamics and control matrices, either using learned representation spaces or hand-crafted ones, and then use the obtained matrices to solve control problems. In another line of work, Watter et al. (2015) seek to identify a time-dependent locally linear model of a system's dynamics in a learned embedding space. However, their approach does not generalize to globally linear models. In all these approaches, the control task is decoupled from the representation space construction, potentially leading to a mismatch between the representation space and the real state space. To address this issue, we propose to include a control task in the learning framework in order to learn a dynamical system representation which is forced to be controllable. In particular, we leverage differentiable optimization techniques from Amos et al. (2018) and Roulet & Harchaoui (2022) in order to include a differentiable LQR controller in the learning framework.

## 3 RESULTS

| | Single pendulum dataset | | | Multiple pendulums dataset | | |
|---|---|---|---|---|---|---|
| | $\alpha = 5°$ | $\alpha = 10°$ | $\alpha = 15°$ | $\alpha = 5°$ | $\alpha = 10°$ | $\alpha = 15°$ |
| $T_c = 30$ | 95 | 100 | 100 | 55 | 84 | 95 |
| $T_c = 35$ | 38 | 80 | 92 | 21 | 40 | 61 |
| $T_c = 40$ | 7 | 17 | 36 | 7 | 15 | 26 |
| $T_c = 45$ | 3 | 6 | 12 | 3 | 6 | 12 |

Table 1: **Success rates of pendulum control** given various thresholds and different control horizons $T_c$. An experiment is considered a success when the pole reaches its target position within a range of +-$\alpha$ degrees. Success rates are expressed in % and were evaluated on 1000 control tasks.

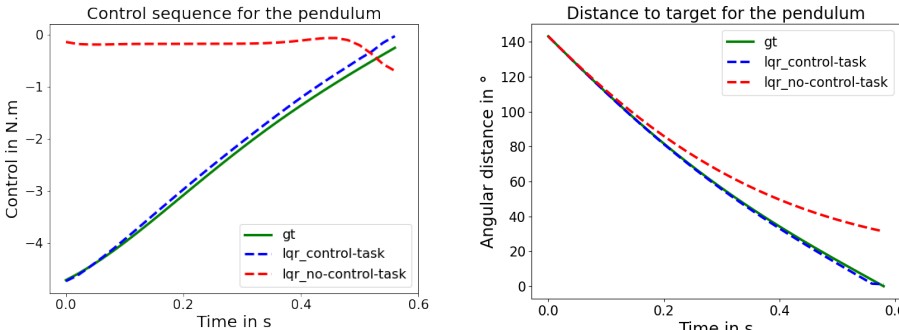

Figure 2: LQR problem of taking a pendulum from an initial position to a target position in 0.6 s (or 30 time steps.). **Left**: control trajectories in N.m. **Right**: distance to target angular position in degrees.

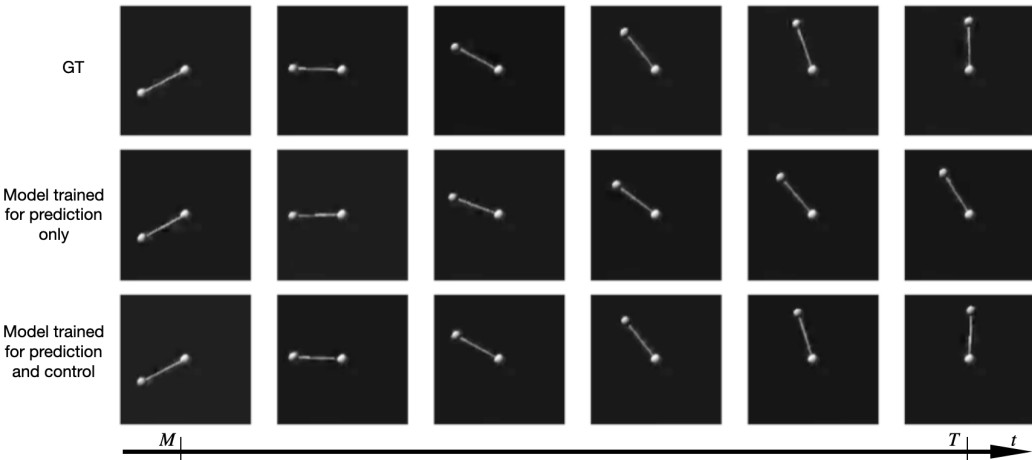

Figure 3: **Pendulum.** Video rendering of the trajectory of the controlled system. In the first row, the system is controlled with the ground truth controls. In the second row, it is controlled with control solutions to the LQR problem using the model trained for prediction only. In the last row, it is controlled using control solutions to the LQR problem using the model trained for prediction and control. All sequences are subsampled (we show 1 frame over 5).

## 3.1 DATASETS

We generate datasets of optimal trajectories for a pendulum and a cartpole. For each system, we specify initial and terminal conditions. We use Pinocchio (Carpentier et al., 2015–2021) to simulate geometric models of the systems, and Crocoddyl (Mastalli et al., 2020) to solve the optimal control problem of taking the systems from the initial condition to the terminal condition. For the pendulum, the initial and terminal conditions are initial and target angular positions and velocities for the pole. Initial positions are sampled uniformly between $\frac{3\pi}{4}$ and $\frac{5\pi}{4}$, and target positions are sampled uniformly between $\frac{-\pi}{4}$ and $\frac{\pi}{4}$. We choose such distributions of initial and target positions to ensure the optimal control trajectories we obtain are not trivial. The cartpole we consider in our experiments is a cart translating along a horizontal axis. A pole is attached to it, and it can rotate over an axis orthogonal to both itself and the cart axis. For the cartpole, the initial and terminal configurations are initial and target Cartesian positions and velocities for the tip of the pole. For all trajectories, the target position of the cartpole is one where the pole is up and vertical. Both datasets contain different trajectories of pendulums and cartpoles with different physical parameters. For the pendulums, the pole length is sampled uniformly between 0.5 m and 0.8 m, and the mass is sampled uniformly between 0.5 kg and 2 kg. For the cartpoles, the carts are cylinders of radius 0.1 m and of

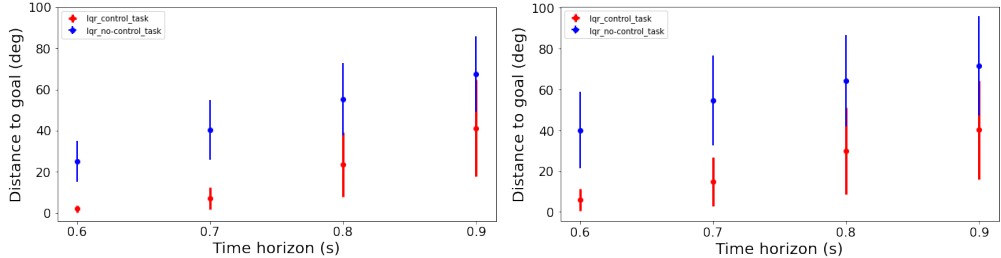

Figure 4: **Pendulum.** Average distance to target as a function of the control horizon. **Left.** Model trained on multiple trajectories of a single pendulum. **Right.** Model trained on multiple trajectories of pendulums with various masses and pole lengths.

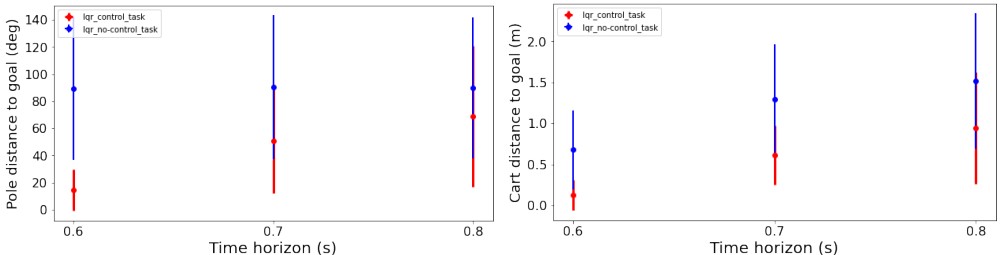

Figure 5: **Cartpole.** Average distance to target as a function of the control horizon. **Left.** Pole distance to target. **Right.** Cart distance to target.

length and mass sampled uniformly respectively between 0.3 m and 0.7 m and 2 kg and 5 kg. The pole length is fixed at 0.7 m, and its mass is sampled uniformly between 0.5 kg and 2 kg. Then, each trajectory is generated from different initial and target conditions. We render both generated systems using panda3d-viewer (Kalevatykh, 2019) into black and white videos of frames of $64 \times 64$ pixels. We train our models on 4000 videos of 10 seconds (200 frames), and test them on 1000 videos of the same duration. Systems are different in both sets: their physical parameters are diverse, and initial and target configurations are random. Thus trajectories in both sets are different.

## 3.2 ARCHITECTURE

The image and control autoencoders are the only functions with trainable parameters in our proposed model (the dynamics matrices $A$ and $B$ are latent variables). The image autoencoder is made of a 6 convolutional blocks encoder, where each convolutional block except for the last one is composed by a $3 \times 3$ convolution layer, a max-pooling layer, a batch normalization layer and a ReLU layer. The last block does not have a batch normalization layer nor a RELU layer. The decoder is symmetric to the encoder (with transposed convolutional layers instead of convolutional layers). The encoder of the control autoencoder is made of a dense layer followed by a tangent hyperbolic (tanh) non-linearity. The decoder is also symmetric to the encoder. We learn one unique autoencoder for all the pendulum systems, and another unique autoencoder for all the cartpole systems. We use the differentiable LQR solver introduced in Amos et al. (2018), to which we add proximal regularization terms to ensure stability during training, similarly to Bounou et al. (2021) for the estimation stage. The LQR cost matrices $Q$ and $R$ are set to the identity matrix. We train our models during 200 epochs on a single Tesla V100 GPU.

## 3.3 LEARNING DYNAMICS WITH A CONTROL TASK IN THE OBJECTIVE

Given measurements of a system performing an optimal trajectory and the associated optimal controls, we learn a representation space where the system dynamics are forced to be linear, and exploit this linearity to solve LQR control problems in this space. Solutions of the LQR problem are then used to control the original system. Since the trajectories are optimal in the measurements space (by construction during the datasets generation), we also look for optimal control trajectories in the

representation space. In our case, the measurements are videos, and the overall task is supervised in the following way: given sequences of length $T$ and a time index $M$ smaller than $T$, codes $z_t$ and $c_t$ from 1 to $M$ are encodings of the measurements $d_t$ and the controls $u_t$ from 1 to $M$: $z_t = \varphi(d_t)$ and $c_t = \psi(u_t)$. We then compute the codes $z_t$ from $M + 1$ to $T$ in two ways. First, we use the forward linear model $z_{t+1} = Az_t + B\psi(u_t)$ using the known $u_t$ that are encoded, and starting from $z_M = \varphi(d_M)$. This is to ensure the encoders $\varphi, \psi$ and the matrices $(A, B)$ are suited for prediction. Next, we solve the LQR problem associated with the task of taking the system from $M$ to $T$ (in $T - M$) time steps. This is to ensure the encoders $\varphi, \psi$ and the matrices $(A, B)$ are suited for control. The definition of codes from $M + 1$ to $T$ is consistent in both ways: first, the codes verify the dynamics constraints by construction; second, they are solution of the LQR problem and are thus feasible (i.e., they also verify the dynamics constraints).

### 3.3.1 CONTROL RESULTS

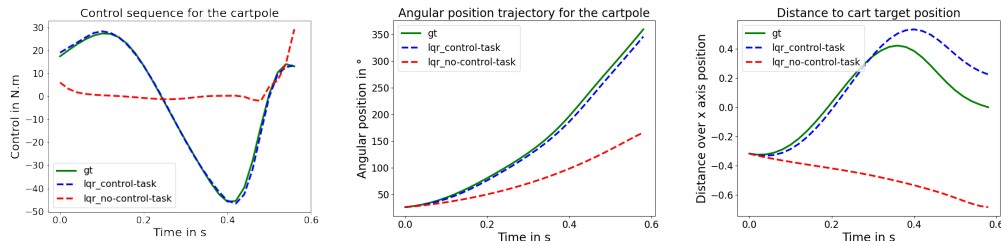

Figure 6: LQR problem of taking a cartpole system from an initial position to a target position in 0.6s (or 30 time steps.). **Left**: control trajectories. **Middle**: Angular position of the pole over time. **Right**: Cart distance to its target position over time.

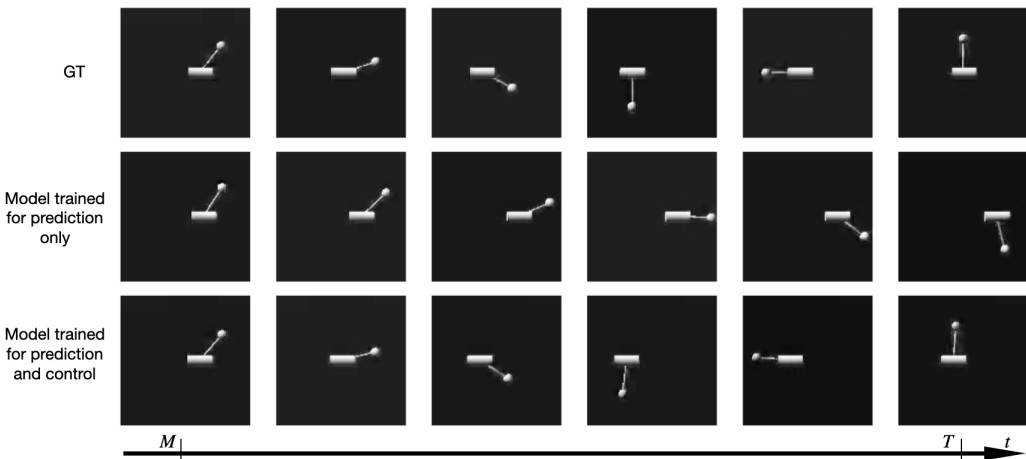

Figure 7: **Cartpole.** Video rendering of the trajectory of the controlled system. In the first row, the system is controlled with the ground truth controls. In the second row, it is controlled with control solutions to the LQR problem using the model trained for prediction only. In the last row, it is controlled using control solutions to the LQR problem using the model trained for prediction and control. All sequences are subsampled (we show 1 frame over 5).

Figure 2 (respectively Figure 6) shows control trajectories (left) and the resulting controlled system trajectories (right). The control trajectories are obtained by solving the LQR problem of taking a pendulum (respectively a cartpole) from an initial position ($t = M$) to a target one ($t = T$) in 30 time steps (0.6 s). The initial and target positions are directly specified in the form of images, enabling image-based control. The ground truth controls (in green solid lines) correspond to the optimal controls in the original space. The LQR controls (in dashed-lines, both blue and red) are obtained by solving the LQR control problem of Eq. (8) in the encoding space, then decoding the

obtained sequence back to the measurements space with the learned controls decoder $\Psi$. The blue dashed-lines correspond to controls obtained with a model that is trained for both future prediction and control (lqr-control-task), while the red dashed-lines correspond to controls obtained with a model that is trained for future prediction only (lqr-no-control-task). We see that in the case where the model is trained for both prediction and control (lqr-control-task), the decoded controls are very close to the ground truth ones. In addition, these decoded controls are effective in driving the systems to their target position. This is not the case anymore with the model trained for prediction only, for which the decoded controls are very different from the ground truth ones, and consequently fail in driving the systems to their target position. A qualitative example of the effectiveness of including the control task in the learning framework can be seen in Fig. 3, and it is all the more noticeable in the case of the cartpole in Fig. 7, which is not surprising since its dynamics are more chaotic than the pendulum's. Figure 4 shows the average over 1000 samples of the distance to the target angular position (in degrees) as a function of the control horizon (in s) for the pendulum. In the left plot, the model is trained on trajectories of one single pendulum, whereas in the right plot, it is trained on trajectories of pendulums with various physical parameters. In both cases, including the control task in the learning framework (lqr-control-task in red) leads to a significant improvement of the control performance. We also report success rates of the models trained with a control task in table 1. The model trained (and evaluated) on a dataset with trajectories of a single pendulum has higher success rates than the model trained (and evaluated) on a dataset with trajectories from multiple pendulums, but the latter still achieves satisfactory results, especially on short control horizons. Figure 5 shows the average over 1000 samples of the distance to target pole and cart positions for the cartpole. We show the average distance of the pole angle in the left plot, and the average distance of the cart position in the right plot. The appraoch trained without a control taks (lqr-no-control-task) completely fails when applied to the cartpole, especially on longer control horizons.

## 4 CONCLUSION

In this work, we introduce an approach to jointly learn and control the dynamics of a nonlinear dynamical system from raw observations directly. We notably leverage the Koopman formalism to learn an encoding to a space where the system dynamics are linear. To enforce the controllabilty of the system in this learned representation space, we include a control task in our learning framework to ensure the learned representation is adapted to control. Experimentally, we show that this joint strategy is effective in controlling complex dynamical systems such as pendulums and cartpoles directly from raw images. In particular, we show that separating the dynamics learning process from the control task leads to infeasible solutions in practice, which are avoided with the combined strategy. In future work, we plan to apply our approach to more complex dynamical systems such as robotic arms to control their movement from raw visual inputs directly, using images to specify target objectives. In addition, we plan to leverage self-supervised approaches to avoid using optimal trajectories examples during training.

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
