# OpenReview forum: "Learning System Dynamics from Sensory Input under Optimal Control Principles"
_ICLR.cc/2023/Conference — Submitted to ICLR 2023_

### Official Review · Reviewer_1Vdd · 2022-10-23

**Confidence:** 5
**Correctness:** 2
**Technical Novelty And Significance:** 2
**Empirical Novelty And Significance:** 2
**Recommendation:** 3

**Clarity, Quality, Novelty And Reproducibility:**

the paper is very clear written. the originality is doubtful. the numerical experiments are not conclusive.

**Strength And Weaknesses:**

1. I do not understand the rational of this work. If you have optimal control, u, you have a solution. Why to calculate it again via LQR?

2. I do not see any connection to "the Koopman formalism" except for the fact that the latent dynamics is represented by matrices (linear dynamics), and the Koopman operator is a linear operator.

3. What is the original objective? Eq.11 is a sample based approximation of the original objective.

4.  Eq.11 combines terms with different units and scales. Are the measurement loss and control loss
on the same scale? The later is in units of actions. The former is in different unites.

5. Directly relevant works (e.g,. Embed to Control: A Locally Linear Latent Dynamics Model for Control from Raw Images Manuel Watter, Jost Tobias Springenberg, Joschka Boedecker, Martin Riedmiller) are not mentioned.





**Summary Of The Paper:**

this paper suggests to simultaneously learn state representation from images and to fit the LQR solution in this state representation (and linear latent dynamics) to the externally provided optimal control signal.

**Summary Of The Review:**

The concerns in "Strength And Weaknesses" prevent me from recommending this paper for publication in ICLR.

---

### Official Review · Reviewer_72mB · 2022-10-24

**Confidence:** 4
**Correctness:** 4
**Technical Novelty And Significance:** 3
**Empirical Novelty And Significance:** 3
**Recommendation:** 5

**Clarity, Quality, Novelty And Reproducibility:**

The paper is well written and clear up until the experiments section, but afterwards some of the notation and clarifying discussion is missing.

* Is the set U a subset of R^p or R^d in eq. (1)?
* Are there any issues in coming up with a good LQR formulation with delayed coordinates? What is the choice of h in the experiments? How did you come up with it? Are there any difficulties, or is it straightforward?
* What is N and M in your experiments? N = 4000? M = ?
* What is the size of the control input space (d ?) Are the controls encoded to a very high dimensional space such that control is linear there? Or is it the opposite?
* Does the dataset generation use LQR to generate optimal examples? How can we avoid feeding optimal examples for more complex systems where it may not be feasible?
* Section 3.3 "real" is misleading. The real system is not "real", as in "real robot experiments" in your case I think.
* In order to make space for more discussion one could move section 3.3.2 to an appendix, it doesn't seem too important for the paper but is presented at the very end.


**Strength And Weaknesses:**

Strengths:

* The paper is well written and motivated, and the presented approach makes sense. The authors show some successful examples where their contribution leads to successful control,

Weaknesses:

* The paper however suffers from lack of extensive experimental support. The authors only show sample sets of images and a few plots where the method leads to an improvement. The effect on the overall test data (1000 videos of 10 seconds each on both the pendulum and the cartpole systems) is not mentioned, nor is it quantified (have I missed it???)

* Some comparative discussion is missing. How much of the improvement is due to the added two cost functions coming from LQR, and how much of it is due to the choice of the network and other details? Didn't the previous Koopman-theory-inspired approaches train autoencoders that led to successful control? As pendulum is the simplest nonlinear system I assume there were successful pendulum toss-ups reported previously. What was the difference there?

* Ablation studies are missing. What is the effect of the the networks chosen, and is it difficult to train? I assume looking at eq. (11) that training with it successful controllers must be quite difficult and prone to issues (local minima, poor performance, nonconvergence etc.)


**Summary Of The Paper:**

This paper presents a latent dynamics learning (or system identification) scenario from very high dimensional observations (images). Inspired by the Koopman operator theory, the authors try to find a latent space where the dynamics is linear. and use this dynamics to control two simple dynamical systems (pendulum toss-up and cartpole) with LQR. Unlike other Koopman-theory inspired approaches, the authors include the (LQR) control problem inside the training scenario and show examples where not including results in failure to control the system successfully.

**Summary Of The Review:**

In general I believe this paper could be accepted, however the experimental evidence is not extensive. I would be willing to increase my score if the authors provide more discussions and quantify their approach more.

---

### Official Review · Reviewer_kRhs · 2022-10-24

**Confidence:** 4
**Correctness:** 2
**Technical Novelty And Significance:** 2
**Empirical Novelty And Significance:** 2
**Recommendation:** 3

**Clarity, Quality, Novelty And Reproducibility:**

The paper is clearly written and the writing quality is good.

The novelty would be satisfactory if the claims were supported with a stronger experimental validation. A purely qualitative evaluation based on a single performed control trajectory for both environments is weak and the major issue I have with the paper.

I rate the Reproducability as good enough. All important equations and techniques used seem to be described in the paper. An available open-source implementation would make it great.

**Strength And Weaknesses:**

Strength:
- The paper is clear, conscise and easy to understand.
- The authors propose two novel changes to koopman-based learning of system dynamics: First, they couple the prediction and control task and learn a representation that has to satisfy both. Second, the authors remove the requirement of the original controls being linear in latent space. These are valuable contributions.

Weaknesses:
1.  The experimental validation seems incomplete and purely qualitative. The authors do not compare against any prior method. Even though these prior methods do not train on the control task directly, they also test the control task directly or learn an actor-critic policy on top of the learned dynamics. DCKNet for example also tests the cartpole. I am missing a quantitative comparison to prior approaches.
The authors also only show a simple successful control trajectory. I would like to see a success rate when randomizing the initial condition of the system.
2. The chosen control strategy, namely LQR, seems problematic for the control problems tested against. Both the pendulum and cartpole are typically considered difficult due to their inherent underactuation. In both cases the pole cannot simply be put into the upright configuration without performing a complex swingup motion. To the best of my knowledge LQR should be incapable of solving such a control problem. In the two control trajectories the authors show in their paper the pole directly moves to the upright configuration.
Therefore, I would like the authors to clarify if the presented pendulum is underactuated. The shown control trajectory for the cartpole starts in an initial configuration which seems to have enough energy such that a direct swingup is possible. Therefore I would like to see the experiment starting from randomized initial conditions with a success rate.
3. The approach the authors propose requires "expert trajectories" because the representation is learned such that the LQR controller in latent space mimics an optimal controller. For more complex systems gathering "expert trajectories" is difficult. Could the authors comment on how this impacts the scalability of their approach to more complex environments.

**Summary Of The Paper:**

The authors propose a method of learning system dynamics from raw sensory input based on Koopman operator theory. Both state and control variables are embedded into a latent space with an encoder-decoder structure. The system dynamics in latent space are forced to be linear. Apart from loss terms measuring the encoding and prediction capabilities, the authors add two losses measuring how well the latent dynamics are controllable using an LQR controller. This is novel compared to prior work.
The approach is validated on two simulated systems: pendulum and cartpole. The authors compare their proposed method to a baseline without the additional control-related loss terms and show qualitatively that their method can control both environments, whereas the baseline fails.

**Summary Of The Review:**

Due to the weak experimental support and my concerns regarding the chosen control approach I think the paper should be rejected. However, I welcome the authors add additional quantitative evaluations to support their claims.

---

### Comment · Reviewer_72mB · 2022-12-01
**My score stays the same**

Dear reviewers,

As the authors supplied us with a detailed and well-written rebuttal, I was obliged to re-read the paper and I spent some time digesting the rebuttal. I think the paper could be accepted but the issues highlighted by the reviewers are still there, hence I will keep my score the same. I am happy to discuss further if needed.

Some final notes:

* Many thanks for the detailed rebuttal. I think if you address the remaining issues, the paper will be a good contribution to the literature.
* I am not convinced that solving (11) is easy given that there are many criterions to consider and many network parameters to optimize (2 autoencoder pairs) together with the extended latent A, B parameters (coming from eq. 13). The authors only mention the difficulty of solving the equality-constrained LQR law, but this issue I believe is rather the least problematic one [I would recommend them to look into solving the Riccati equations more stably, there is many papers in the controls literature addressing this issue, as an obvious first thing to check for instance, the Riccati matrix P needs to be symmetric positive definite at each iteration]. For the next iterations of the paper, I would recommend authors to add ablation studies and more discussion as to the difficulties of solving (11) to find a "stable" feedback law.
* Now that the authors added Table I, stability of the learned feedback law seems to be an issue. I suggest the authors extend the formulation to find a stable feedback law (that will keep the state of the system close to the desired state for a much longer horizon).

---

### Decision · Program_Chairs · 2023-01-20

**Decision:**

Reject

**Justification For Why Not Higher Score:**

See meta-review.

**Justification For Why Not Lower Score:**

n/a

**Metareview: Summary, Strengths And Weaknesses:**

The paper introduces learning system dynamics from sensory input based using linear operators.  Apart from lacking references to existing literature on locally linear latent spaces, the paper is not convincing in its experimental validation; also, ablation studies are lacking.  The choice of LQR seems suboptimal.

It is suggested that the many points raised by the reviewers---who mostly favor a reject---are addressed, including using MPC, and then go for a submission at a future venue.